# Glutamine Supplementation as an Anticancer Strategy: A Potential Therapeutic Alternative to the Convention

**DOI:** 10.3390/cancers16051057

**Published:** 2024-03-05

**Authors:** Hayato Muranaka, Rasaq Akinsola, Sandrine Billet, Stephen J. Pandol, Andrew E. Hendifar, Neil A. Bhowmick, Jun Gong

**Affiliations:** 1Department of Medicine, Cedars-Sinai Medical Center, Los Angeles, CA 90048, USA; hayato.muranaka@cshs.org (H.M.); rasaq.akinsola@cshs.org (R.A.); sandrine.billet@cshs.org (S.B.); stephen.pandol@cshs.org (S.J.P.); andrew.hendifar@cshs.org (A.E.H.); neil.bhowmick@cshs.org (N.A.B.); 2Samuel Oschin Comprehensive Cancer Institute, Cedars-Sinai Medical Center, Los Angeles, CA 90048, USA; 3Department of Biomedical Sciences, Cedars-Sinai Medical Center, Los Angeles, CA 90048, USA; 4Department of Research, VA Greater Los Angeles Healthcare System, Los Angeles, CA 90073, USA

**Keywords:** cancer, metabolism, glutamine, amino acids, nutrition, cancer therapy, cachexia

## Abstract

**Simple Summary:**

Glutamine, vital for the body’s functions, is pivotal in cancer metabolism as it influences tumor growth. However, cancer cells’ complex adaptive metabolic dynamics raise concerns about potential limitations in glutamine antagonism strategies to impede tumor growth. Similarly, while glutamine supplementation shows promise in supporting cancer patients, careful considerations are necessary to address possible interactions with ongoing treatments and concerns about inadvertent tumor growth stimulation. Recent studies have shed light on the effects of glutamine on the epigenetic regulation of cancer cells and the enhancement of anti-cancer immune functions, providing valuable insights for potential therapeutic advancements. Understanding the intricacies and challenges of glutamine interventions is essential for optimizing their potential benefits in cancer treatment and patient well-being.

**Abstract:**

Glutamine, a multifaceted nonessential/conditionally essential amino acid integral to cellular metabolism and immune function, holds pivotal importance in the landscape of cancer therapy. This review delves into the intricate dynamics surrounding both glutamine antagonism strategies and glutamine supplementation within the context of cancer treatment, emphasizing the critical role of glutamine metabolism in cancer progression and therapy. Glutamine antagonism, aiming to disrupt tumor growth by targeting critical metabolic pathways, is challenged by the adaptive nature of cancer cells and the complex metabolic microenvironment, potentially compromising its therapeutic efficacy. In contrast, glutamine supplementation supports immune function, improves gut integrity, alleviates treatment-related toxicities, and improves patient well-being. Moreover, recent studies highlighted its contributions to epigenetic regulation within cancer cells and its potential to bolster anti-cancer immune functions. However, glutamine implementation necessitates careful consideration of potential interactions with ongoing treatment regimens and the delicate equilibrium between supporting normal cellular function and promoting tumorigenesis. By critically assessing the implications of both glutamine antagonism strategies and glutamine supplementation, this review aims to offer comprehensive insights into potential therapeutic strategies targeting glutamine metabolism for effective cancer management.

## 1. Introduction

Cancer therapy continues to face challenges arising from treatment-related toxicities and the intricate metabolic adaptations of cancer cells, necessitating a comprehensive understanding of the roles of cancer metabolism in both tumor progression and patient well-being. Amino acids play immeasurable roles in cell survival through nucleotide biosynthesis, redox balance management, epigenetic regulation, and immune response associated with tumorigenesis and metastasis [1]. Many studies have highlighted amino acids essential for tumor growth, including glutamine, proline, and serine [2,3]. Glutamine (Gln), a “conditionally essential” amino acid with diverse roles in cellular physiology [4,5], has earned significant attention in cancer research and therapy [6,7]. Due to its implication in various cellular processes, ranging from energy production to nucleotide biosynthesis, glutamine has been of growing interest in cancer biology research, where it has been shown to support the aberrant growth and survival mechanisms of cancer cells [7,8,9]. Antagonism of Gln metabolism has been studied as a foundational approach for targeting cancer metabolism by blocking glutamine transporters or glutaminase [6,10,11,12]. However, these efforts do not provide durable benefits due to multiple mechanisms [12,13]. Therefore, several studies have helped unravel the interplay between glutamine metabolism and how to tailor this toward cancer treatment strategies effectively [12,14].

In recent years, research has attempted to show the power of glutamine modulation as a complementary approach in cancer therapy [11,15,16]. Glutamine supplementation offers a promising potential for improving treatment-related toxicities by fortifying immune function and possibly enhancing the overall well-being of cancer patients [5]. The versatile benefits of glutamine supplementation include its role in promoting gut integrity, mitigating mucositis, and supporting the maintenance of intestinal barrier function in the face of aggressive cancer treatments [17,18]. Nevertheless, glutamine supplementation in cancer therapy is not without challenges. This includes optimal dosages, potential interactions with standard treatments, tailored approaches for specific cancer subtypes, and adequate patient cohort choices, which are crucial considerations. The complex interplay between glutamine metabolism and cancer progression necessitates a nuanced understanding of the inherent implications of glutamine supplementation on tumor growth, emphasizing the need for comprehensive clinical investigations to delineate its precise role in personalized cancer care [10,19].

This review offers broad insights into the evolving landscape of glutamine precise modulation as a potential adjunct therapy in cancer treatment while underscoring the significance of personalized approaches in harnessing the full potential of this amino acid in cancer therapy.

## 2. Regulation of Glutamine Metabolism in Cancer

Regulation of glutamine metabolism in cancer cells involves complex molecular and signaling pathways that dynamically influence cellular processes, including proliferation, survival, and adaptive responses to tumor microenvironments [20]. Central to this regulatory network are vital enzymes such as glutaminase (GLS), glutamine synthetase (GS), and glutamine transporters (SLC1A5, also known as ASCT2, etc.), whose activities are finely modulated by various oncogenic and tumor suppressive pathways involving primary metabolites, such as glutamate (Glu), α-ketoglutarate (α-KG), and lactate. These crosstalk help establish a metabolic landscape favoring increased glutamine uptake and utilization in cancer cells [21]. Dysregulated glutamine metabolism in cancer cells facilitates the synthesis of macromolecules (proteins, nucleotides, and lipids), oxidative stress/redox homeostasis, epigenetic functions, and anabolic processes crucial for sustaining cancer cells’ rapid proliferation and survival [22]. On the other hand, cancer cells also exhibit a significantly high consumption of glucose (the Warburg Effect) and lactate production. Like glucose, glutamine can be degraded to lactate and produce the nicotinamide adenine dinucleotide phosphate (NADPH) as a by-product of this flux via malic enzyme, contributing to fatty acid synthesis by facilitating the cell’s ability to use glucose-derived carbon [6,8,23,24]. However, glutamine metabolism reprogramming often occurs under nutrient-limited conditions and metabolic stress, highlighting the capacity of the cancer cell to adapt by rewiring its metabolism pathways to its environment and making glutamine a vital metabolic substrate. Regulation and roles of glutamine metabolism in cancer cells are depicted in Figure 1.

Among the prominent signaling pathways influencing glutamine metabolism [25], the K-RAS pathway is a pivotal driver of glutamine dependency in various cancer types [9,26]. Mutations in the K-RAS oncogene are commonly found in pancreatic, colorectal, and lung cancers, where it has been shown to regulate glutamine uptake (glutamine addiction) and subsequent utilization of anabolic processes unfolding non-canonical pathways crucial for sustaining rapid tumor cell proliferation and growth [27,28]. The MYC oncogene, known for its critical role in cellular growth and proliferation, profoundly impacts glutamine metabolism by amplifying the expression of glutamine transporters and glutaminase [29]. MYC promotes the conversion of glutamine to glutamate to fuel the TCA cycle and facilitate anabolic pathways essential for cellular division and growth. The MYC-driven metabolic program highlights the indispensable role of glutamine as a key player in the continuous proliferation associated with several cancer types [30]. Additionally, dysregulation in the phosphoinositide 3-kinase (PI3K)/AKT/mammalian target of rapamycin (mTOR) pathway further accentuates the demand for glutamine, driving an accelerated flux through the tricarboxylic acid (TCA) cycle to fuel the biosynthetic requirements of rapidly proliferating cancer cells [31].

Similarly, the tumor suppressor protein p53, known for its multifaceted roles in maintaining genomic stability and orchestrating cellular stress responses, exerts intricate control over glutamine metabolism [32]. In the face of genotoxic stress, p53 directs a metabolic shift that reduces glutamine consumption, thus promoting the conservation of cellular resources and redirecting metabolic intermediates toward pathways essential for cellular survival [33]. Moreover, p53 suppresses the expression of glutamine transporters and glutaminase, curbing glutamine utilization and reshaping the metabolic mercenaries to favor stress adaptation and cellular survival. The dynamic interplay between p53 and glutamine metabolism underscores the delicate balance between tumor cell metabolism and the intricate signaling pathways that govern cellular responses to the tumor microenvironment [34]. The retinoblastoma (Rb) protein, another critical tumor suppressor, also modulates glutamine metabolism in the context of cancer. Rb regulates the expression of genes involved in cell cycle progression and metabolism, impacting the utilization of glutamine for various biosynthetic pathways necessary for cell growth and proliferation [35]. Perturbations in the Rb signaling axis can lead to alterations in glutamine metabolism, promoting the metabolic adaptations required for sustained tumor growth and progression.

Moreover, glutamine metabolism is also affected by the tumor microenvironment (TME) (Figure 2). Several studies reported that tumors, including pancreatic ductal adenocarcinoma (PDAC) and esophageal cancer, inherently have low glutamine levels compared to benign adjacent tissues or other normal tissues [36,37,38]. This may induce different metabolic reprogramming, including activating macropinocytosis and other alternative metabolic pathways, to meet nutrient demands under glutamine-restricted conditions [39,40,41,42]. Furthermore, tumor core regions display low glutamine levels compared to the periphery, and this regional glutamine deficiency in tumors promotes de-differentiation through the inhibition of histone demethylation [40,43]. Additionally, the competition for glutamine between tumor cells and immune cells in the TME causes glutamine deficiency, affecting immune cells’ function [44]. Interestingly, cancer-associated fibroblasts (CAFs) also display heightened macropinocytosis induced by glutamine starvation or oncogenic signaling, providing a source of intracellular amino acids and secreted amino acids that support CAF functions and tumor cell survival [45,46].

In summary, regulating glutamine metabolism in cancer cells involves a complex interplay of diverse oncogenic and tumor-suppressive pathways and the complex influence of TME, underscoring glutamine’s dynamic and context-dependent roles in supporting tumorigenesis and malignant progression. These oncogenic and tumor-suppressive pathways also play a pivotal role in regulating glycolysis [47,48,49,50,51]. Thus, both glutaminolysis and glycolysis contribute synergistically to fuel the heightened energy demands of cancer cells, provide essential building blocks for biomass synthesis, and foster the uncontrolled growth characteristic of malignancies. A comprehensive understanding of these regulatory mechanisms is crucial for developing targeted therapeutic strategies to disrupt the glutamine-dependent metabolic needs of cancer cells and impede their proliferative potential.

## 3. Glutamine Deprivation Strategies in Cancer

Over the last few decades, glutamine has become an attractive cellular target for cancer therapy due to its pleiotropic roles in fundamental cellular functions. The complexity and adaptability of cancer metabolism have prompted considerable interest in targeting glutamine metabolism as a therapeutic strategy. Glutamine metabolism has become a focal point for intervention in cancer due to its role in sustaining the increased metabolic demands of rapidly proliferating malignant cells [11]. Several approaches for depriving cancer cells of glutamine have been attempted, with encouraging preclinical data, although clinical translation has yet to be formally achieved. Among those strategies are glutamine mimicking compounds (DON and JHU083), glutamine transporter blockers (GPNA, V-9302, and JPH203), major glutamine-producing enzymes (CB-839 and BPTES), glutamine depletion (L-asparaginase), and multiple downstream actors of glutamine biosynthesis [12,14,52].

Among the more recent strategies targeting glutamine metabolism for cancer therapy is the inhibition of glutaminase. Glutaminase (GLS) is an enzyme that catalyzes the conversion of glutamine to glutamate, a crucial step in glutamine metabolism [53,54]. Glutaminase inhibitors, such as CB-839, have shown promise in preclinical studies and early-phase clinical trials, demonstrating their ability to impede glutamine catabolism and disrupt the supply of glutamate, a precursor for numerous biosynthetic pathways critical for cancer cell survival and proliferation [12,55]. The disruption of the vital enzymatic steps associated with glutamine utilization by glutaminase inhibitors results in reduced availability of metabolic intermediates required for nucleotide synthesis, tricarboxylic acid (TCA) cycle replenishment, and redox balance maintenance, collectively inducing metabolic stress and compromising cancer cell viability [56]. In addition to the effects on tumor cells, CB-839 may improve immune function in a tumor microenvironment by enhancing the anti-tumor activity of both autologous T-cell therapies and checkpoint inhibitor therapies in a mouse melanoma model [57].

Another avenue for glutamine metabolism-targeted interventions lies in modulating the transport systems responsible for glutamine uptake by cancer cells [58]. Glutamine is transported into and out of cancer cells via specific transporters, such as SLC1A5 (ASCT2) and SLC7A5 (LAT1), which play integral roles in sustaining the increasing glutamine demands of proliferating cancer cells [59,60]. Inhibitors targeting these transporters, such as V-9302 and JPH203, have emerged as potential strategies for disrupting the influx of extracellular glutamine, thereby impairing its supply to intracellular metabolic pathways [61,62]. This inhibition strategy induces cellular stress by impeding anabolic processes and undermining the adaptability of cancer cells to the dynamic tumor microenvironment [63]. Notably, the expression and functionality of glutamine transporters and other genes/proteins related to glutamine metabolism may differ significantly between cancer vs benign cells and different types of cancers [55,64]. Additionally, even within the same type of cancer, there can be substantial heterogeneity in terms of glutamine reliance based on genotype or subtype of cancer [65,66]. Understanding the contrasting expression patterns and functional aspects of these glutamine metabolism-related molecules is crucial for comprehending the broader implications of targeted therapies, including insights into potential side effects associated with therapeutic strategies, thereby enhancing the precision and safety of cancer treatments.

Beyond glutaminase inhibition and modulation of transport systems, other targeted therapeutic approaches are being explored to disrupt glutamine metabolism in cancer cells. For instance, targeting glutamine synthetase (GS), the enzyme that catalyzes the conversion of glutamate to glutamine, represents an alternative strategy to perturb glutamine homeostasis within cancer cells [67]. In preclinical studies, inhibition of glutamine synthetase has demonstrated anti-proliferative effects by disrupting glutamine-dependent pathways crucial for cancer cell survival [68]. Additionally, efforts are underway to potentially tailor therapeutics to be able to distinguish metabolic tumor status, for example, by intervening in glutamine utilization in specific subsets of cancer cells that exhibit glutamine addiction.

Despite the promise of glutamine deprivation strategies, several challenges and limitations warrant consideration [13]. As discussed earlier, some cancers show low glutamine levels compared to benign adjacent tissues [36,37,38], raising a question about the rationale of this therapeutic approach (Figure 2). Several studies have reported significantly decreased serum glutamine levels in cancer patients compared to healthy individuals, possibly due to a higher glutamine demand by cancer cells [65,69]. Of note, low serum glutamine levels were associated with advanced-stage, higher proinflammatory cytokine levels, poorer overall survival (OS), and progression-free survival (PFS) compared with those with high glutamine levels in colorectal cancer (CRC) patients [70]. Furthermore, cancer cells often display remarkable metabolic adaptability, allowing them to reroute metabolic flux through alternative pathways in response to therapeutic interventions. This phenomenon, known as metabolic plasticity, can undermine the efficacy of glutamine-targeted strategies, resulting in the need to comprehensively understand and predict cancer cells’ reprogramming mechanisms. For example, PDAC cells scavenge extracellular proteins by oncogenic KRAS-driven macropinocytosis to adapt to nutrient-restricted microenvironments such as glutamine-restricted conditions [39,40]. Additionally, cancer cells can reprogram their metabolism to utilize, if necessary, other carbon sources for survival, such as asparagine and aspartate [41,42]. The intricate metabolic microenvironment within the tumor, characterized by nutrient gradients, hypoxia, and interactions with stromal cells, adds another layer of complexity to the design and implementation of glutamine-targeted therapies [40,43,44,45,46].

Some attempts at blocking glutamine metabolism in cancer patients resulted in unacceptable toxicity, particularly in the gastrointestinal tract [71]. Leone et al. designed a prodrug form (JHU083) of the glutamine antagonist DON, administered in an inert state but then preferentially activated by enzymes enriched in the tumor microenvironment [72]. Glutamine blockage by JHU083 downregulated glycolysis and oxidative phosphorylation in cancer cells, resulting in a concomitant decline in hypoxia, acidosis, and nutrient depletion while enhancing anticancer immunity and oxidative phosphorylation in T-cells. These findings suggest exploiting cellular metabolic plasticity to modulate T-cell metabolism and antitumor immune responses through glutamine blockade.

In conclusion, the evolving landscape of glutamine deprivation strategies in cancer therapy highlights the diverse approaches to disrupting glutamine metabolism in specific vital cellular nodes. The pursuit of precision medicine in this context involves not only elucidating the intricacies of glutamine-dependent pathways but also addressing the challenges posed by metabolic plasticity and the complex evolving tumor microenvironment. Despite these challenges, the ongoing research in this field holds promise for advancing therapeutic strategies that exploit the vulnerabilities of cancer cells dependent on glutamine for their survival and proliferation (Figure 3).

## 4. Glutamine Supplementation in Cancer

Glutamine has emerged as a potential adjunct therapy in the comprehensive management of cancer. Beyond its fundamental role as a building block for protein synthesis, glutamine has garnered significant attention for its diverse functions in supporting immune function, preserving gut integrity, and alleviating treatment-related toxicities, thereby improving the overall well-being and quality of life of cancer patients undergoing various therapeutic regimens [5,18,73].

The immune system is vital in surveilling and eliminating cancer cells, which significantly rely on glutamine to sustain their metabolic demands [5]. Glutamine aids in maintaining the redox balance within immune cells, thereby preventing oxidative stress-induced damage and promoting their longevity and functionality. Several studies have highlighted the potential of glutamine supplementation in enhancing the anti-cancer immune response, presenting a compelling avenue for improving the efficacy of immunotherapeutic approaches in cancer treatment [73]. In an early experimental study, Klimberg’s group revealed that glutamine gavage (1 g/kg/day) reduced tumor growth by 40%, associated with a 30% increase in natural killer (NK) cell activity in tumor-bearing animals [74]. A recent study demonstrated that directly enhancing intratumoral glutamine abundance affects anti-tumor immunity in subcutaneous tumor xenograft in immunocompetent mice [75]. Glutamine supplementation inhibited tumor growth by enhancing type-1 conventional dendritic cells (cDC1s)-mediated CD8+ T cell immunity, overcoming resistance to immunotherapies. Mechanistically, tumor cells and cDC1s compete for glutamine uptake, and glutamine signaling via folliculin (FLCN) affects nutrient- and stress-sensitive transcription factor TFEB function. Interestingly, enhanced antitumor responses were also observed by glutamine blockade, and this was associated with an increased glutamine level in the tumors [72].

Glutamine supplementation has demonstrated a significant impact on preserving gut integrity, particularly in cancer therapy-associated gastrointestinal toxicities. Cancer treatments, including chemotherapy and radiation therapy, often exert detrimental effects on the gastrointestinal tract, leading to mucosal damage (mucositis), compromised barrier function, and intestinal inflammation [76]. Glutamine is known for its trophic effects on the intestinal mucosa by helping maintain the structural integrity of the gut lining and supporting the regeneration of damaged mucosal cells [77]. By promoting the proliferation of intestinal epithelial cells and enhancing mucin synthesis, glutamine supplementation has shown promise in alleviating gastrointestinal toxicities and reducing the severity of treatment-induced side effects. This helps strengthen the tolerability of anticancer therapies and improves patients’ overall quality of life [78]. Additionally, by acting as a precursor for glutathione, an essential antioxidant, glutamine aids in mitigating oxidative stress and reducing the risk of chemotherapy-induced neurotoxicity, such as peripheral neuropathy [79].

Furthermore, its role in promoting protein synthesis, reducing inflammation, and preserving muscle mass has garnered attention in managing cancer cachexia. Cancer cachexia is a debilitating condition characterized by increased inflammation resulting in progressive muscle wasting and weight loss [80]. Glutamine supplementation has shown promise in alleviating the severity of cachexia by improving the nutritional status and functional capacity of cancer patients undergoing intensive treatment regimens [81]. Glutamine also contributes to maintaining acid-base balance by its ability to produce ammonium ions, a typical weak base [82]. Since acidosis is a hallmark of inflammatory processes as well as a crucial determinant of tumor progression, resulting from the increased glycolysis and the subsequent accumulation of lactic acid as well as the massive production of protons [83], glutamine supplementation may help to reduce inflammation in cancer cachexia patients by maintaining pH homeostasis in the body via ammonia (NH_3_) transport between tissues [5,84]. Furthermore, glutamine can change the composition or function of microbiota [78] by potentially influencing gut microbial metabolism, including the production of the metabolites from the tumor-promoting/suppressing gut microbiome, metabolites that contribute to extracellular acidosis, and short-chain fatty acids (SCFAs) that provide energy to colonic epithelial cells [85,86,87,88].

Recent studies have also highlighted the potential of glutamine supplementation in modulating epigenetic regulation in cancer cells [43,89]. Glutamine serves as a critical substrate for α-ketoglutarate (α-KG) production, which is an essential cofactor for several enzymes involved in epigenetic modifications, e.g., Jumonji C (JmjC)-containing histone lysine demethylases (KDM) and DNA demethylases ten–eleven translocation (TET) enzymes [90,91]. In melanoma tumors, low glutamine levels often drive the cells to an undifferentiated state; however, upon supplementation with dietary glutamine, tumor growth was significantly inhibited, pushing the tumor to become sensitive to a BRAF inhibitor. Mechanistically, it was revealed that glutamine uptake effectively increased the concentration of tumor glutamine and its downstream metabolite α-KG, which drove hypomethylation of H3K4me3 via JmjC-containing KDMs, resulting in the downregulation of epigenetically activated oncogenic pathways [89]. By influencing the epigenetic landscape of cancer cells, glutamine supplementation holds promise in regulating gene expression patterns and modulating critical signaling pathways implicated in tumorigenesis and cancer progression. Additionally, understanding the interplay between glutamine metabolism and epigenetic regulation offers a novel perspective for leveraging glutamine supplementation as an adjunct therapy to conventional cancer treatments, potentially enhancing their efficacy and improving patient outcomes.

Glutamine is directly or indirectly responsible for the uptake and efflux of other amino acids, such as alanine (Ala), threonine (Thr), leucine (Leu), and cystine (Cys), via transporters such as SLC1A5 (ASCT2), SLC38A1, SLC6A14, SLC7A5 (LAT1), and SLC7A11 (xCT). Based on a competitive inhibition mechanism, adding supraphysiological concentrations of glutamine (SPG) inhibited the uptake of neutral amino acids in oocytes expressing human SLC1A5 (ASCT2), the primary glutamine transporter in cancer cells [92,93]. Interestingly, anti-cancer effects of increased intratumoral glutamine were observed in subcutaneous mouse tumor xenograft models where L-Gln was directly injected into tumors at a lower dose than that prescribed for sickle cell disease patients [43,75,94]. While it has been considered challenging to manipulate local glutamine concentrations in vivo, especially in the tissues by oral supplementation or enteral feedings, the typical methods for glutamine supplementation [5], this finding highlights the therapeutic importance of local intratumoral glutamine levels for cancer patients. Extended investigations are warranted to clarify the effects of SPG on amino acid uptake in cancer cells and the response to chemotherapy drugs.

Overall, the diverse roles of glutamine supplementation in cancer therapy, encompassing immune modulation, gut integrity preservation, mitigation of treatment-related toxicities, and its potential impact on epigenetic regulation and amino acid transport, underscores the promising therapeutic potential of this amino acid in the comprehensive management of cancer. Integrating glutamine supplementation into the existing treatment paradigm can improve the tolerability and efficacy of conventional cancer therapies and enhance patients’ overall well-being and quality of life during their cancer treatment journey. However, further research is warranted to elucidate the optimal dosing regimens, possible interferences with other therapeutic modalities, and long-term safety profile of glutamine supplementation in different cancer patient populations (Figure 4).

## 5. Clinical Evidence of Glutamine Modulation

Clinically targeting glutamine metabolism has been pushed forward following cancer metabolism’s exceptional plasticity and conventional therapies’ limits (recurrence and resistance) [38]. Several drugs have been developed with promising animal studies; nevertheless, only limited ones have been attempted in clinical trials [11]. Recently, a GLS1 selective inhibitor, CB-839 (telaglenastat), tested in preclinical trials showed no significant side effects and, therefore, was moved forward into clinical trials. While the in vitro model using KRAS-derived PDAC cells had a promising impact, the in vivo experiment, unfortunately, did not achieve reduced cell growth due to the adaptive metabolomic plasticity of those cells in response to CB-839 [43]. Accordingly, CB-839 has been tested on solid tumors in a combination trial. In these clinical studies, CB-839 has been administered with nivolumab as a treatment for melanoma, renal cell carcinoma (RCC), and non-small cell lung cancer (NSCLC); everolimus for RCC; palbociclib for KRAS-derived PDAC, NSCLC and CRC; and cabozantinib for advanced RCC. In vitro and animal studies have shown that CB-839 can potentially improve the radiosensitivity of head and neck carcinomas (HNSCC) and NSCLC, making it a good candidate for chemotherapy and radiotherapy in clinical settings.

The phase 1 clinical trials evaluated escalating doses of CB-839 in patients with advanced solid tumors among the patients with triple-negative breast cancer, RCC, mesothelioma, NSCLC, fumarate hydratase (FH)-deficient tumors, succinate dehydrogenase (SDH)-deficient gastrointestinal stromal tumors (GIST) or non-GIST tumors, cMyc-mutated tumors, and isocitrate dehydrogenase-1 (IDH1)- or IDH2-mutant tumors in combination with standard of care chemotherapy. Some adverse clinical effects were observed, such as diarrhea, decreased appetite, nausea, and fatigue. However, the trial outcome was encouraging, with good patient tolerability [95].

On the other side of the spectrum, the only FDA-approved glutamine therapy is in sickle cell disease (SCD) patients of all ages based on a Phase III clinical trial where L-glutamine was well tolerated and reduced pain crisis [96]. There exists no formal FDA approval of glutamine supplementation as a cancer therapeutic, however. Instead, the bulk of the clinical data that supports glutamine supplementation to date exist in clinical studies where glutamine supplementation was used as an adjunct to chemotherapy or radiation therapy to mitigate toxicities from cancer therapies (Table 1) [73].

Glutamine supplementation has garnered significant attention for its potential to bolster immune function, a critical aspect of the body’s defense against cancer [5]. Studies in patients with esophageal cancer on radiochemotherapy reported that high-dose oral glutamine supplementation (30 g/day) can restore the lymphocyte count and enhance lymphocyte mitogenic function. This suggests a positive impact on immune function during cancer treatment [97,98]. While further research is required to delineate the specific conditions under which glutamine supplementation may exert maximal immunomodulatory effects in different cancer types, these immune-enhancing effects hold promise for optimizing the host’s defense mechanisms against cancer cells and potentially enhancing the efficacy of immunotherapeutic interventions.

Chemotherapy-induced mucositis, a debilitating side effect characterized by inflammation and ulceration of the mucous membranes, has been a particular target in glutamine supplementation studies [18]. Findings from clinical trials exploring oral or parenteral glutamine supplementation in cancer patients undergoing chemotherapy have presented a mixed picture [73]. While some studies suggest a potential reduction in the severity and duration of mucositis with glutamine supplementation, others report more modest or inconclusive outcomes [99,100,101,102,103]. In several studies, glutamine supplementation, both oral (including “swish and swallow” therapy) and parenteral, has effectively reduced the severity and duration of stomatitis in patients receiving chemotherapy [104,105]. It is unclear whether glutamine can ameliorate diarrhea associated with abdominal radiation or chemotherapy; studies showed mixed results [106,107,108]. Moreover, it also showed promising results in preventing intestinal absorption and permeability impairments during chemotherapy [97,98,107,109,110,111]. Variability in study protocols, glutamine dosages, and patient cohorts may contribute to the divergent findings, emphasizing the need for standardized approaches, and larger, well-controlled trials to elucidate the true impact of glutamine supplementation on mucositis and other treatment-related toxicities.

The potential benefits of glutamine supplementation extend beyond immune modulation and enteric toxicity mitigation to encompass broader aspects of patient well-being during cancer treatment [73]. Studies examining the impact of glutamine on nutritional status, quality of life, and treatment tolerance have provided insights into the multifaceted effects of this amino acid supplementation [112,113,114,115,116,117]. The positive outcomes of these clinical studies with different types of cancer patients/treatments include a reduction in cancer cachexia-related protein, an improvement in protein synthesis, a significant improvement in fat-free mass and serum albumin, maintenance of the lean body, and prevention of radiation-induced injury and body weight loss. Maintaining adequate nutritional status in cancer patients undergoing intensive treatments is crucial for sustaining energy levels, preserving muscle mass, and supporting overall health [118]. Glutamine, as a key substrate for protein synthesis and an essential component of the body’s antioxidant defenses, has been explored in both experimental and clinical studies for its potential to diminish treatment-induced weight loss and improve nutritional parameters. Clinical evidence, however, presents a complex narrative, with some studies suggesting benefits in weight maintenance and improved quality of life, while others report more equivocal outcomes. The influence of factors such as treatment regimens, cancer types, and patient-specific characteristics should be further examined to define the contexts in which glutamine supplementation may be most advantageous for the overall well-being of cancer patients.

## 6. Perspective, Challenges, and Future Directions

Despite the promising roles of glutamine deprivation strategies and supplementation in cancer therapy, several challenges persist, including the need for a comprehensive understanding of the metabolic adaptation of cancer cells, personalized delivery methods, and identifying patient subgroups likely to benefit the most (Figure 3 and Figure 4). The fine-tuning between giving L-glutamine and depriving it is still not well understood and is dependent on a specific personalized patient’s profile and timing. Moreover, the debate between localized, stable glutamine levels, and circulating levels within the tumor and the surrounding cells is of significant concern and needs thorough dissection and mechanistic understanding (Figure 2). Addressing these challenges requires continued research to optimize the clinical application, minimize potential side effects, enhance the efficacy of these interventions, and develop improved glutamine-like chemicals or better combination strategies.

The safety profile of glutamine supplementation is critical to its clinical evaluation. While glutamine is generally considered safe, there are some adverse effects reported in clinical studies as limited/rare adverse events, including toxicity in the liver and kidneys [119,120,121]. Additionally, concerns regarding potential interactions with specific cancer types and treatments persist mainly due to the early in vitro knowledge that cancer cells preferably consume glutamine (Figure 4). Several studies have provided evidence suggesting that glutamine is not linked to the promotion of tumor growth and does not adversely affect the outcomes of diverse cancer treatments. In a study involving breast cancer patients, the reduction in tumor size did not show a significant difference between the groups receiving glutamine and those receiving a placebo [107].

Similarly, a study by Topkan et al. found that oral glutamine supplementation in patients with locally advanced non-small cell lung cancer undergoing chemoradiotherapy did not lead to significantly different outcomes in terms of cancer-related clinical results, overall survival, and progression-free survival compared to the group without glutamine treatment [122]. Another investigation by Tsujimoto et al. focused on patients with head and neck cancer, revealing no significant difference in the overall response rate, which represents the percentage of patients achieving complete or partial response, between the glutamine and placebo groups 10 weeks post-chemoradiotherapy completion [123]. However, high doses of glutamine supplementation in certain contexts, particularly in patients with gastrointestinal cancers, have raised questions about the theoretical risk of fueling cancer growth. We are currently investigating the safety and feasibility of a combination of gemcitabine and nab-paclitaxel with oral L-glutamine powder as a first-line systemic therapy in patients with unresectable or metastatic PDAC (GlutaPanc phase I trial [124]). This will be the first-ever prospective trial to employ a dose-finding design of an approved oral L-glutamine therapy along with first-line standard chemotherapy in untreated metastatic PDAC. The intricate balance between providing a substrate for normal cellular functions and potentially supporting malignant cells underscores the importance of careful consideration and individualized approaches in the clinical application of glutamine supplementation.

Clinical evidence for the optimal delivery methods, concentration, and chemical stability of glutamine supplementation remains an area of ongoing exploration; the efficacy of the treatment can be influenced by factors such as solubility and absorption. Glutamine solubility is low (25 g/L); thus, suspensions are needed for topical, oral, and enteral supplementation; therefore, adding disaccharides can facilitate mucosal uptake. Manipulating glutamine levels is challenging because glutamine is abundant in the body, as studies show minimal changes in plasma glutamine levels even after repeated high-dose supplementations [125]. Additionally, monitoring local glutamine concentrations might be difficult since, in general, there is a poor correlation between plasma concentration and tissue concentrations [126]. Moreover, whether administered orally, intravenously, or even locally, the bioavailability and effectiveness of glutamine supplementation may vary. Determining the optimal dosage and duration of supplementation and identifying patient populations that derive the most benefit are crucial considerations for refining the clinical application of glutamine supplementation in cancer therapy. In addition, glutamine delivery through free and dipeptide forms has been explored, focusing on the efficacy of glutamine dipeptides [127]. Glutamine dipeptides, particularly l-alanyl-l-glutamine (Ala-Gln), have shown efficacy in reducing infectious complications, hospital stay length, and mortality in critically ill patients, as supported by clinical and experimental studies [127,128,129,130,131,132,133,134]. The choice between free glutamine and glutamine dipeptides depends on the patient’s catabolic circumstance and route of administration. Therefore, it is crucial to consider patient-specific factors in deciding the route, dose, and form of glutamine supplementation.

## 7. Conclusions

Glutamine, as a key player in cancer metabolism, holds significant promise as a target for therapeutic intervention. While both glutamine deprivation strategies and supplementation offer potential benefits in cancer treatment, their complex interplay with tumor microenvironments and metabolic adaptations necessitates further research and refinement to maximize clinical efficacy and ensure patient safety. As we navigate the intricacies of glutamine modulation, sustained efforts in research and development remain imperative to harness its full therapeutic potential in the fight against cancer.

## Figures and Tables

**Figure 1 cancers-16-01057-f001:**
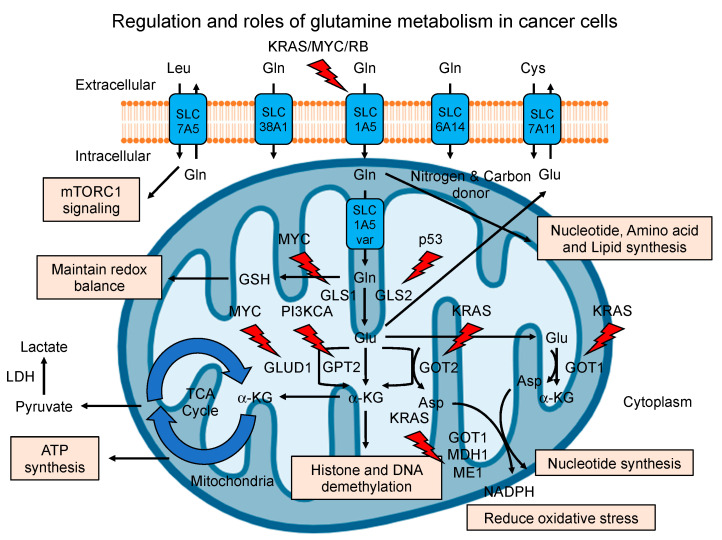
Regulation and roles of glutamine metabolism in cancer. Cancer cells take up glutamine (Gln) through glutamine transporters, including SLC1A5 (ASCT2), SLC38A1, and SLC6A14. For glutaminolysis, glutamine is transported into the mitochondrial matrix through the SLC1A5 variant and subsequently catalyzed to glutamate (Glu) by glutaminases (GLS1 and GLS2). Glutamine and glutamate are indirectly responsible for the uptake of other amino acids, such as leucine (Leu) and cystine (Cys), via the SLC7A5 (LAT1) and SLC7A11 (xCT) transporters, respectively. Glutamate is converted to α-KG through GLUD1 or aminotransferases (GPT2, GOT1 and GOT2). α-KG contributes to epigenetic modifications by histones and DNA demethylation. The resulting intermediates can supply bioenergetics through the TCA cycle and support the biosynthesis of proteins, nucleotides, and lipids. In addition, glutamine metabolism maintains redox balance via GSH synthesis and reduces oxidative stress via NADPH synthesis. Glutamine metabolism in cancer is regulated by oncogenes (KRAS, MYC, and PI3KCA, etc.) and tumor suppressor genes (p53 and RB, etc.) (red lightning).

**Figure 2 cancers-16-01057-f002:**
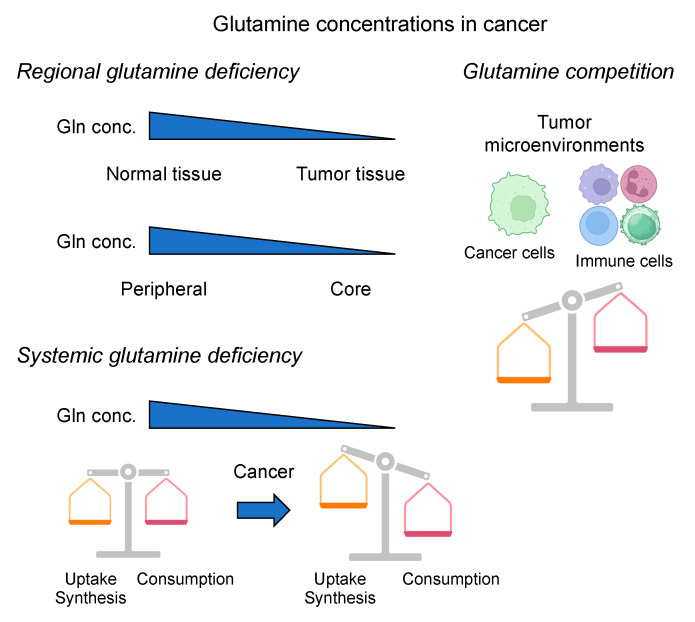
Glutamine concentrations in cancer. **Upper left**: Regional glutamine deficiency. Tumors inherently have low glutamine levels compared to benign adjacent tissues. Furthermore, tumor core regions display low glutamine levels compared to the periphery, and this regional glutamine deficiency in tumors promotes de-differentiation through the inhibition of histone demethylation. **Upper right**: the competition for glutamine between tumor cells and immune cells in the tumor microenvironment (TME) causes glutamine deficiency, affecting immune cells’ function. **Lower left**: systemic glutamine deficiency. Blood glutamine concentration changes according to the balance between significant organ producers (liver, skeletal muscle, lung, adipocytes, etc.) and consumers (brain, kidney, gut, liver, immune cells, etc.) in health and catabolic situations. In health, there is a balance between glutamine synthesis and degradation. In contrast, under stress and/or catabolic conditions such as cancer, organs responsible for glutamine synthesis (such as the skeletal muscle tissue) reduce its production, and at the same time, immune cells increase their demand for glutamine. Cancer patients have significantly decreased serum glutamine levels compared to healthy individuals; under this condition, the endogenous synthesis of glutamine does not appear to meet the human body’s demand, and glutamine assumes the role of a conditionally essential amino acid.

**Figure 3 cancers-16-01057-f003:**
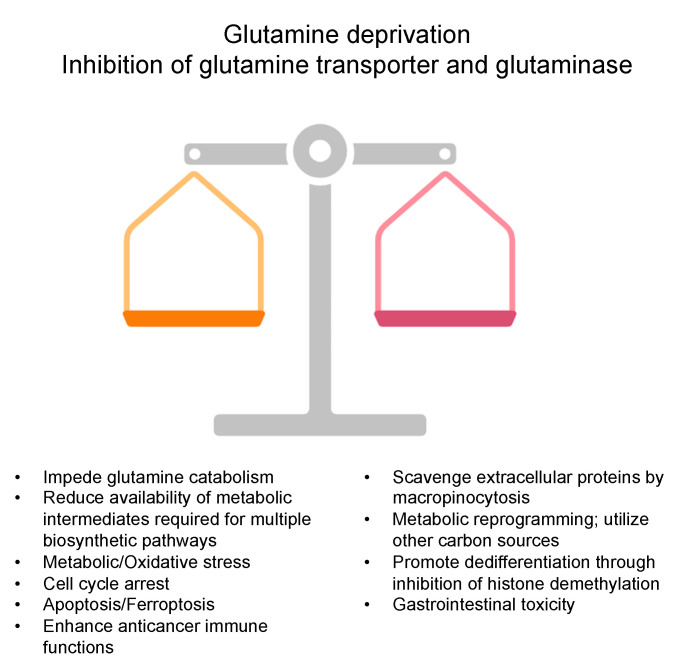
Benefits and drawbacks of targeting glutaminolysis in cancer.

**Figure 4 cancers-16-01057-f004:**
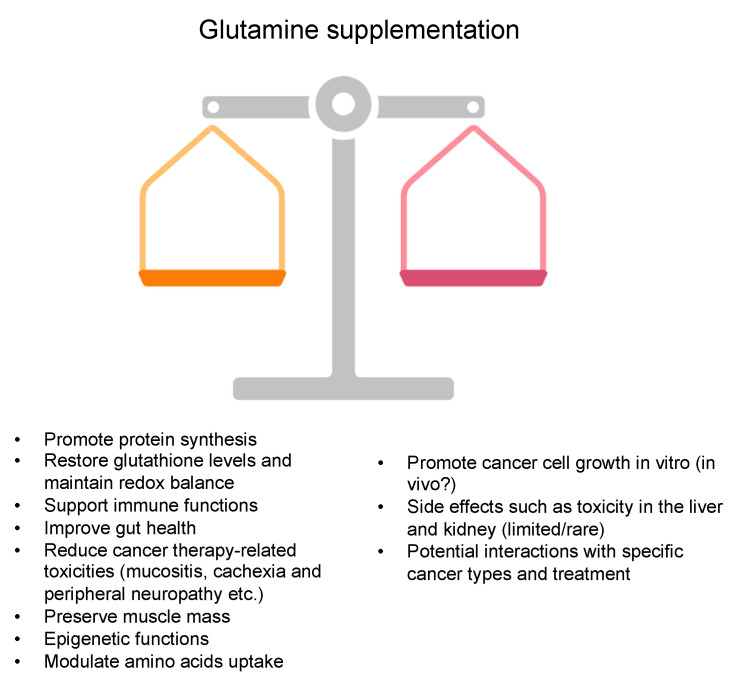
Benefits and drawbacks of glutamine supplementation in cancer.

**Table 1 cancers-16-01057-t001:** Clinical trials of glutamine supplementation in cancer patients.

Patients (Cancer Types)	*n*	Therapy	Gln Supplementation	Outcome	Ref.
Esophageal cancer	13	Radiochemotherapy; cisplatin and 5-FU *^1^	30 g/day Gln over 28 days	Reduction in the lymphocyte count prevented Gut permeability attenuated	[97]
Esophageal cancer	13	Radiochemotherapy	30 g/day Gln over 4 weeks	Enhanced lymphocyte mitogenic function Reduced gut permeability	[98]
Gastrointestinal cancer	28	Chemotherapy; 5-FU and FA *^2^	16 g/day Gln for 8 days	Good tolerance No significant effect on oral mucositis	[99]
Metastatic colorectal cancer	24	Chemotherapy; 5-FU and CF *^3^	0.4 g/day i.v. Gly-Gln for 5 days	Reduction in mucositis, gastric ulcerations and duodenal mucosa	[100]
Head and neck cancer	17	Radiation therapy	2 g/m^2^ Gln swish therapy, 4×/day	Shorter duration and severity of oral mucositis	[101]
Breast cancer	326	Chemotherapy; cyclophosphamide *^4^, doxorubicin and 5-FU	2.5 g Gln (Saforis), 3×/day for 14 days	Reduced incidence of ≥ grade 2 and 3 mucositis Safe and effective	[102]
Metastatic colorectal cancer	86	Chemotherapy; oxaliplatin	15 g Gln, 2×/day for 7 days, every 2 weeks during chemotherapy	Lower incidence of oxaliplatin-induced mucositis No effect on response to chemotherapy and survival	[103]
Soft tissue sarcoma, osteosarcoma, Kaposi’s sarcoma, and breast cancer	14	Chemotherapy; doxorubicin, dacarbazine, CP, etc.	4 g Gln swish and swallow 2×/day	Decreased severity and number of days of mucositis	[104]
Sarcoma and neuroblastoma	24	Chemotherapy; 5-FU and leucovorin; carboplatin and etoposide, methotrexate	4 g/day Gln for at least 14 days	Reduction in duration and severity of chemotherapy-associated stomatitis and mouth pain	[105]
Breast cancer	65	Doxifluridine and leucovorin	30 g/day Gln for 8 days	Doxifluridine-induced diarrhea not prevented No impact on tumor response to chemotherapy	[106]
Breast cancer	60	Chemotherapy	Gln ≥ 12 days	Amelioration of the chemotherapy-induced increase in intestinal permeability No significant positive effects on stomatitis, diarrhea, and the antitumor effect of chemotherapy	[107]
Rectal, bladder, prostate, and gynecologic cancers and pelvic soft tissue sarcomas	36	Radiotherapy	15 g Gln, 3×/day for 2 weeks	No difference in overall diarrhea incidence None of the patients in the glutamine-treated group had grade 3–4 diarrhea, but in the placebo group, grade 3–4 diarrhea was seen in 69% of the patients	[108]
Advanced/metastatic colorectal cancer	70	Chemotherapy; 5-FU and FA	18 g/day Gln for 15 days	Reduction in intestinal absorption and permeability	[109]
Gastrointestinal cancer	39	Chemotherapy; CF and 5-FU	30 g/day Gln for 7 days	Decrease in intestinal permeability	[110]
Advanced/metastatic cancer	51	Chemotherapy; 5-FU and leucovorin	30 g/day Gln for 15 days	Lower intestinal permeability score with Gln vs. controls Lower incidence of grade 2–4 mucositis	[111]
Breast cancer	32	HMB *^5^	14 g/day of Gln for 24 weeks	Reduction in cancer cachexia-related protein Improvement in protein synthesis	[112]
Hematological, solid cancer, and multiple sclerosis	40	High-dose chemotherapy with autologous stem cell transplantation	30 g/day Gln for 14 days	Causes proteo-catabolism of medium severity	[113]
Neck and head malignancy	44	Surgery	0.3 g/day Gln for 4 weeks	Significant improvement in fat-free mass and serum albumin Maintenance of lean body	[114]
NSCLC *^6^	60	Concurrent radiotherapy	10 g/8 h Gln for 12 months	Radiation-induced injury and body weight loss prevented	[115]
Gastric adenocarcinoma	1950	Gastrectomy	0.05–0.49 g/kg/day Gln	A decrease in serum albumin levels	[116]
Lung cancer (SCLC*^7^ or NSCLC)	96	-	2.4 g HMB, 14 g Arg *^8^, Gln 14 g Gln/day for 12 weeks	A patient lived without significant loss of lean body mass	[117]

*^1^ 5-FU, 5-fluorouracil; *^2^ FA, folinic acid; *^3^ CF, calcium folinate; *^4^ CP, cyclophosphamide; *^5^ HMB, β-hydroxy-β-methylbutyrate; *^6^ NSCLC, non-small cell lung cancer; *^7^ SCLC, small cell lung cancer; *^8^ Arg, arginine.

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
