# Peer review of "Glutamine Supplementation as an Anticancer Strategy: A Potential Therapeutic Alternative to the Convention"

_cancers, 2024, doi:10.3390/cancers16051057_

Round 1

Reviewer 1 Report

Comments and Suggestions for Authors

See attached file

Author Response

1) The manuscript is well written and quite coherent, with appropriate references for the discussed topic. I do find the “debate” perspective interesting with regards to utilizing glutamine blockade therapy as well as glutamine supplementation in cancer patients, which would seem counterintuitive at first. This review shows sufficient critical analysis of the current state of literature, offering perspectives for future studies in the field.

We are grateful for the overwhelmingly positive comments.

2) Though being relatively well documented, I do question the originality of the current review given alternative reviews (doi: 10.3390/nu12061675 or 10.1038/s12276-023-00971-9 or 10.1093/immadv/ltab010) already summarize extensive knowledge of the current state of literature in either glutamine blockade or glutamine supplementation in cancer therapy.

We understand that glutamine metabolism has been extensively studied in the cancer research field, and there are a lot of excellent reviews. However, there are gaps in the comprehensive understanding of glutamine blockade and glutamine supplementation strategies in cancer therapy. With the recent update in glutamine supplementation in cancer therapy, including our original preclinical/clinical studies, we tried to bring the above aspects to activate the discussion in the field further.

3) I have to comment on whether the title is appropriate: suggesting that glutamine supplementation is an anticancer strategy might be overstated, given that only a limited number of papers currently describe potential anticancer mechanisms. Rather, an important proportion of the reviewed literature suggests that the role of glutamine supplementation in cancers relies on the management of adverse events linked to traditional chemotherapies and radiotherapy. Also, the title does not mention that this review discusses glutamine metabolism blockade, which encompasses an entire section (3).

We understand the reviewer’s suggestion about the title. As the reviewer mentioned, we are well aware of the issues: our review discusses both glutamine metabolism blockade and glutamine supplementation in cancer, the majority of studies about glutamine supplementation in cancer reported its role in the management of adverse events caused by chemotherapy and radiotherapy, and only a limited number of papers described potential anticancer mechanisms of glutamine supplementation. As stated in the title “A potential therapeutic alternative from convention,” we discussed these conventional strategies. We tried to comprehensively clarify the problems, concerns, and potential alternative approaches. Therefore, our focus remains on the later parts. That said, we would like to keep the title as it is.

4) I believe that a lot of the discussion within this review is quite broad, not delving deeply into mechanisms explaining the utility of glutamine supplementation for cancer patients.

o There is some generalization with regards to glutamine metabolism in tumors; the authors should highlight the importance of heterogeneity between tumor types as well as tumors of the same type concerning their metabolic activity as well as the relative importance of glutamine.

This is a crucial aspect to be discussed in our review. We added some sentences and references about the expression/functionality of glutamine metabolism-related genes/proteins, including the plasma membrane transporters in different/same types of cancer (lines 206-217).

o Certain statements show too much certainty that have not been exhaustively shown/proven by literature: e.g. at lines 56-59 regarding glutamine blockade not being effective in cancers, though only citing a reference studying pancreatic cancer.

We cited the review that comprehensively discusses the mechanisms that induce resistance to glutamine-targeting therapies (line 60). Additionally, we edited the reference in the introduction: the papers about specific cancer types were replaced (lines 53-62).

o The statement at lines 150-153 should include appropriate references to papers investigating such glutamine blockade agents. Currently, this statement is too broad. Readers should be able to have access to the papers discussing these agents.

We added references in the text for the statement on lines 179-182.

o At lines 188-190, the authors state that tumors are relatively depleted in glutamine compared to adjacent tissues, though this is only supported by one publication for pancreatic cancer. How can the authors generalize such a statement to all cancer types with only one appropriate reference for a single tumor type?

In addition to the study on pancreatic cancer, we added references, including the paper about esophageal cancer and the review that cites some more studies in the text (lines 229-230).

o At line 275, reference 66 does not seem to concord or support the mentioned statement on the mechanisms (“luminal pH and other mechanisms”) by which glutamine influences gut microbiota. The authors should refrain from using broad statements such as “other mechanisms” given that they lack important details essential for reader comprehension.

We edited the sentence and added the appropriate reference (lines 327-333).

o The section 3 reports quite general statements regarding glutamine metabolism blockade for cancer therapy, which does not bring much novelty compared to other reviews discussing such phenomena more in detail (10.1038/s12276-023-00971-9).

We understand this point. However, we included and emphasized the problems and potential concerns in this strategy as well as the recent new approaches that have yet to be well discussed in other reviews (lines 228-257), bringing some novelties into our work.

o At lines 243-245 (ref 63), it could be specified, if possible, how anti-tumor immunity is affected by increased intratumoral glutamine, which could offer better insight into two the mechanistic processes linking glutamine supplementation and anti-cancer immune responses.

We added more detailed information in the text (lines 296-300).

o At lines 382-384, references 98 to 103 could be discussed with greater detail.

We added more detailed information in the text (lines 451-455).

5) Discussion of the regulation of glutamine metabolism in cancer is appropriate, though it would be interesting to mention other phenomena contributing to metabolic reprogramming encompassing glutamine within the cancer phenotype: the authors later discuss tumor microenvironment with regards to glutamine availability in the tumoral milieu- how do different TME (e.g. hypoxia) influence glutamine metabolism and dependence in cancer cells? This should be complementary to intrinsic pathways (e.g. oncogenes and TSG) influence glutamine metabolism in cancer.

We appreciate this suggestion since it may improve our manuscript. We added the sentence about how Gln metabolism is affected in TME in section 2 (lines 138-158). How it influences the effects of glutamine blockade therapy is discussed in section 3 (lines 228-248).

6) In section 5, the authors discuss clinical trials with CB-839. Please note that there are additional clinical trials (clinicaltrials.gov) that evaluate the anti-neoplastic efficacy of this agent in cancers. Also, an important proportion of section 5 (table 1) relies on the discussion of relatively outdated references (20-25 years old). Do the authors have any insight as to why glutamine supplementation has not been studied recently alone or in combination with anti-cancer therapies? Nonetheless, the summary in table 1 is clear and concise.

Our list of clinical trials may need to be more comprehensive to cover all information, especially recent unpublished studies. Instead, we picked up the studies based on the importance/impact and outcomes/results. Several excellent reviews comprehensively discuss the clinical trials, such as Eur J Nutr . 2010 Jun;49(4):197-210. doi: 10.1007/s00394-009-0082-2. We pi., and we cited these reviews to avoid overlapping.

7) In figure 2, the panel on the left depicts phenomena not having been well discussed in the review and does not include any pertinent supporting reference. Also, the panel on the right depicts phenomena that are not discussed in the section in which the figure is cited.

We added the sentence about figure 2 in the text and edited figure 2 to match the text.

8) In figure 4, the authors should include that supraphysiological supplementation of glutamine may show toxicity, for example, in patients with liver or kidney disease (e.g. https://doi.org/10.1186/cc13781).

We added this information and reference to figure 4 and the text (lines 481-483).

Reviewer 2 Report

Comments and Suggestions for Authors

This is a well-structured review about glutamine metabolism and transport in cancer cells. The paper is very well written and contains a general introduction, a brief explanation about the metabolism. The paper continues with how glutamine is depleted in cancer, how works with an oral supplementation of glutamine, clinical evidence and the future expectations. In general, I consider that it could be publish in Cancer.

A glutamine supplementation could increase immune function, but also promote cancer growth and therefore is important to know exactly the quantity of glutamine that should be taken orally. The figures with the weighing machine are a impactant to understand this better.

Minor aspects:

Line 309.- “is warranted to clarify”. To is written twice.

Line 281.- It is true that α-ketoglutarate is an inhibitor of mTOR, and thus affecting lifespan and healthspan (https://doi.org/10.1016/j.exger.2023.112154), but I have never read about it is an essential cofactor. Maybe this line should be rewritten.

Line 51.- Glutamine seems to be a conditional essential amino acid in ill conditions (i.e. in cancer). 10.1111/j.1753-4887.1990.tb02967.x  Not very sure that the sentence can be understand by reader, as this is the Introduction. Maybe it should be more extensively explained.

Author Response

1) Line 309.- “is warranted to clarify”. To is written twice.

We corrected it (line 365).

2) Line 281.- It is true that α-ketoglutarate is an inhibitor of mTOR, and thus affecting lifespan and healthspan (https://doi.org/10.1016/j.exger.2023.112154), but I have never read about it is an essential cofactor. Maybe this line should be rewritten.

α-ketoglutarate (α-KG) is an essential cofactor for several enzymes involved in epigenetic modifications, such as Jumonji C (JmjC)-containing histone lysine demethylases (KDM) and DNA demethylases ten–eleven translocation (TET) enzymes. We added the reference and more information in the text (lines 335-339).

3) Line 51.- Glutamine seems to be a conditional essential amino acid in ill conditions (i.e. in cancer). 10.1111/j.1753-4887.1990.tb02967.x  Not very sure that the sentence can be understand by reader, as this is the Introduction. Maybe it should be more extensively explained.

We understand the point. However, to stay focused on the role of glutamine in cancer in this manuscript, we do not explain its general roles in detail, especially in the introduction. Alternatively, we added one reference for the text's sentence (lines 52-53).

Reviewer 3 Report

Comments and Suggestions for Authors

The manuscript by Muranaka et al., describes potential use of glutamine and related metabolism in cancer therapy. Overall it is an interesting topic. I believe that the present work requires some changes to make the issue more understandable without misleading less familiar with the topic readers.

Comments

1. Section 2. When reading this section one gets the impression that cancer cells depend exclusively on glutamine to fuel their metabolism. I believe, this section should be more general and include also some information about the use of glucose by cancer cells. This is particularly necessary as the authors mention K-RAS mutation which are known to redirect cancer metabolism to aerobic glycolysis. I believe it is also necessary to explain how glucose and glutamine is metabolised and how this is regulated/inter-regulated in cancer cells. Preferably with an additional figure.

2. I do not understand the meaning of the sentence - line 94-97. Lactate is normally produced from pyruvate but not by mitochondrial oxidation. Quite the opposite, lactate can be converted into pyruvate and then supply mitochondria. I think authors should make sure that all the statements in this manuscript are correct.

3. There is also something wrong with the sentence - line 154 and this should be corrected.

4. In the section describing potential glutamine-related targets in cancer therapy the authors should mentioned how the targets are expressed/function in benign cells. That would help to understand the side effects of the therapies which are described later in the text. It is important especially regarding expression of the plasma membrane transporters which are mentioned in several places (e.g. line 298) but their role and expression in cancer versus benign cells is not explained. It would also help to understand the potential applicability of glutamine supplementation.

5. Line 150, glutamine transporters blockers are not targets. This sentence needs to be corrected.

Comments on the Quality of English Language

As above

Author Response

1) Section 2. When reading this section one gets the impression that cancer cells depend exclusively on glutamine to fuel their metabolism. I believe, this section should be more general and include also some information about the use of glucose by cancer cells. This is particularly necessary as the authors mention K-RAS mutation which are known to redirect cancer metabolism to aerobic glycolysis. I believe it is also necessary to explain how glucose and glutamine is metabolised and how this is regulated/inter-regulated in cancer cells. Preferably with an additional figure.

We agree with this suggestion, although we didn’t add another figure for that alone. We added some sentences about the interplay between glycolysis and glutamine metabolism in cancer (lines 97-100 and 162-166).

2) I do not understand the meaning of the sentence - line 94-97. Lactate is normally produced from pyruvate but not by mitochondrial oxidation. Quite the opposite, lactate can be converted into pyruvate and then supply mitochondria. I think authors should make sure that all the statements in this manuscript are correct.

Like glucose, glutamine can be degraded to lactate. However, we agree that the sentence needed to be clearer in the context. We edited the sentence and added more references to be easily understood (lines 97-100).

3) There is also something wrong with the sentence - line 154 and this should be corrected.

We corrected the sentence in the manuscript (line 183).

4) In the section describing potential glutamine-related targets in cancer therapy the authors should mentioned how the targets are expressed/function in benign cells. That would help to understand the side effects of the therapies which are described later in the text. It is important especially regarding expression of the plasma membrane transporters which are mentioned in several places (e.g. line 298) but their role and expression in cancer versus benign cells is not explained. It would also help to understand the potential applicability of glutamine supplementation.

This is a crucial aspect. We added some sentences and references about the expression/functionality of the plasma membrane transporters in cancer vs benign and different/same types of cancer (lines 206-217).

5) Line 150, glutamine transporters blockers are not targets. This sentence needs to be corrected.

We corrected the sentence in the manuscript (line 179).